# OpenReview forum: "Robustness Evaluation of Proxy Models against Adversarial Optimization"
_ICLR.cc/2024/Conference — Submitted to ICLR 2024_

### Official Review · Reviewer_8v7i · 2023-11-01

**Soundness:** 3 good
**Presentation:** 2 fair
**Contribution:** 2 fair
**Rating:** 5
**Confidence:** 3

**Summary:**

The paper presents a comprehensive study on the robustness of neural network proxies, specifically in the text domain, under various optimization pressures. The authors introduce "ProxyBench," a benchmark designed to investigate the robustness of these proxies, and conduct extensive experiments to uncover properties of the proxy gaming problem and highlight issues with current proxy reward models used in fine-tuning large language models. The paper emphasizes the importance of robust proxies in machine learning applications across diverse domains and sheds light on the challenges posed by proxy gaming, where proxies may fail to accurately represent the true objective under optimization pressure.

**Strengths:**

1. **Address a Crucial Issue**: The paper targets the significant issue of proxy gaming in machine learning, attempting to bring attention to how proxies may fail to accurately represent true objectives under optimization pressure. Addressing this problem is crucial for the advancement of reliable machine learning applications.

2. **Introduction of a Benchmark**: The authors introduce "ProxyBench," aiming to provide a framework for evaluating the robustness of neural network proxies. This benchmark has the potential to standardize evaluations and comparisons of different models.

**Weaknesses:**

1. **Enhanced Clarity Needed**: The paper is rich in information and analysis, yet there is a noticeable need for enhanced clarity and brevity. The articulation of the problem statement and the methodology is somewhat unclear, making it challenging for readers to fully grasp the proposed works.

2. **Constrained to Text Domain**: The benchmarks and experiments conducted in the paper are exclusively centered around the text domain. This singular focus could potentially limit the broader applicability of the findings and the relevance of the benchmark across diverse domains within the realm of machine learning.

3. **Ambiguous Contributions**: From my perspective, the authors presuppose that the proxy will undergo optimization pressure, and they proceed to evaluate the proxy under this specific condition, using it as a benchmark. However, they fall short of providing a compelling rationale for this assumption. Additionally, the utility of the benchmarks provided remains unclear. On the validation front, the benchmark's credibility is questionable as it is solely based on the performance of one DeBERTa-v3 model, which is insufficient for establishing validity across various model types.

4. **Lack of novelty**: The optimization methods employed in the paper, such as BoN, RL, and white box optimization, are pre-existing techniques. The authors do not offer a justification for their selection of these particular methods for the benchmark, nor do they delve into an analysis comparing the optimization processes of these different methods. This leaves a gap in the paper’s contribution to the field, as it does not provide novel insights or methodologies.

**Questions:**

1. Why do we need an adversarially robust proxy?
2. What are the contributions of the current work?

---

> ### Author Response · Authors · 2023-11-22
> **Author Response (1/2)**
>
> Thank you for your careful analysis of our work. We hope the following response addresses your concerns.
>
> **Clarifying the problem statement.**
>
> We have clarified the problem statement in Section 4 (changes are in blue). Thank you for your suggestion.
>
> **Reason for focusing on text domain.**
>
> > The benchmarks and experiments conducted in the paper are exclusively centered around the text domain. This singular focus could potentially limit the broader applicability of the findings and the relevance of the benchmark across diverse domains within the realm of machine learning.
>
> Our work is about the robustness of reward models to proxy gaming. Currently, reward models are primarily used for fine-tuning LLMs using RLHF or similar methods. These techniques are one of the main ways that people train instruction-following LLMs like ChatGPT, so focusing on the text domain is high-impact. Additionally, the vast majority of research on this topic focuses exclusively on the text domain.
>
> Recently, people have begun to fine-tune multimodal instruction-following models. Research on this topic is still in its infancy, so we felt focusing our benchmark on the text domain was more appropriate. However, our setup could easily be extended to multimodal models in the future when suitable multimodal preference datasets are more widespread.
>
> **Underlying assumption that proxy will undergo optimization pressure.**
>
> > From my perspective, the authors presuppose that the proxy will undergo optimization pressure, and they proceed to evaluate the proxy under this specific condition, using it as a benchmark. However, they fall short of providing a compelling rationale for this assumption.
>
> This is a standard assumption of the proxy gaming problem. Without optimization, proxy gaming wouldn’t occur in the first place. We describe this briefly in the first paragraph of our introduction. For more background, see [1, 2, 3].
>
> **Selection of gold reward models.**
>
> > the benchmark's credibility is questionable as it is solely based on the performance of one DeBERTa-v3 model, which is insufficient for establishing validity across various model types
>
> We use several DeBERTa-v3 models (small, base, and large sizes) as gold reward models. We based our selection of the gold reward models on their test accuracy and efficiency. In preliminary experiments, we evaluated other architectures, including T5-11B and LLaMA models. These obtained only marginally higher accuracy, and in some cases lower accuracy, while being much more expensive to evaluate. Thus, we fixed the gold reward models to the highly performant DeBERTa-v3 architecture, which is a common choice for reward modeling and becoming a standard in this area [4, 5, 6]. If we have addressed the thrust of your concerns, we kindly ask that you consider raising your score.

---

> ### Author Response · Authors · 2023-11-22
> **Author Response (2/2)**
>
> **Sources of novelty and contributions.**
>
> > The optimization methods employed in the paper, such as BoN, RL, and white box optimization, are pre-existing techniques
>
> We do not claim to develop novel optimizers. Rather, we use a range of existing optimizers to study proxy gaming. We may have confused you with the section title “4.2 Constructing Stronger Optimizers”, which was an oversight on our part. Our apologies for this. We have changed this section title to “4.2 Optimization Methods” in the updated paper to improve clarity.
>
> The sources of novelty in our work are as follows.
> We introduce a standardized evaluation methodology and benchmark for proxy gaming in RLHF settings, which was previously not available. We carefully design the evaluation, including the metrics and gold reward models, and we make our code and datasets fully available. (Prior work on this problem did publicly release code or datasets, and did not develop standardized metrics for measuring proxy gaming, so we hope our public benchmark will facilitate future work on this important problem.)
> Contrary to the findings of Gao et al. (2022), we observe larger policies can sometimes cause more severe proxy gaming, demonstrating adversarial effects in the RLHF setting
> We discover the concerning phenomenon of example-specific proxy gaming, which was not previously known.
> Adversarial training is an intuitively plausible method for improving proxy robustness. We adversarially train proxy reward models using text-based optimizers, and we find that this improves proxy robustness against white-box optimizers but not RL or BoN optimizers. This had not previously been explored, and it suggests that new methods will be needed to improve performance on our benchmark.
>
> [1]: “Scalable agent alignment via reward modeling: a research direction”. Jan Leike, David Krueger, Tom Everitt, Miljan Martic, Vishal Maini, Shane Legg. arXiv 2018.
>
> [2]: “Unsolved Problems in ML Safety”. Dan Hendrycks, Nicholas Carlini, John Schulman, Jacob Steinhardt. arXiv 2021.
>
> [3]: “Scaling laws for reward model overoptimization”. Leo Gao, John Schulman, Jacob Hilton. arXiv 2022.
>
> [4]: “Aligning AI With Shared Human Values”. Dan Hendrycks, Collin Burns, Steven Basart, Andrew Critch, Jerry Li, Dawn Song, Jacob Steinhardt. ICLR 2021
>
> [5]: “Safety-Tuned LLaMAs: Lessons From Improving the Safety of Large Language Models that Follow Instructions”. Federico Bianchi, Mirac Suzgun, Giuseppe Attanasio, Paul Röttger, Dan Jurafsky, Tatsunori Hashimoto, James Zou. arXiv 2023
>
> [6]: “Reward Collapse in Aligning Large Language Models”. Ziang Song, Tianle Cai, Jason D. Lee, Weijie J. Su. arXiv 2023

---

### Official Review · Reviewer_tXHZ · 2023-11-05

**Soundness:** 3 good
**Presentation:** 3 good
**Contribution:** 4 excellent
**Rating:** 6
**Confidence:** 3

**Summary:**

The paper introduces ProxyBench, a benchmark tool for assessing the robustness of neural network proxies in the text domain against optimization pressures. It reveals limitations in the effectiveness of current robustness enhancement techniques, like adversarial training, and exposes significant issues with existing proxy reward models. The experiments conducted with ProxyBench indicate a need for new methods to ensure proxy robustness, setting a foundation for future research in this area.

**Strengths:**

The strengths of the work is listed below:
1. The introduction of ProxyBench is a substantial contribution, as it provides a practical and comprehensive benchmark for assessing the robustness of neural network proxies against optimization pressure, especially in the text domain.
2. The paper conducted extensive experimental studies, and the insights from the statistics are very meaningful.
3. This work uncovers the relationship between proxy model size and its robustness, contributing to the understanding of how different factors like model parameters, training data, and training steps affect proxy stability.
4. The introduction of metrics for evaluating proxy reward model robustness and detailed analysis on the performance of baseline methods are practical.
5. The exploration of different optimization strategies, including best-of-n, reinforcement learning, and white-box optimization, indicates a deep understanding of the landscape and provides a nuanced perspective on potential vulnerabilities.

**Weaknesses:**

1. Assumption of Gold Model Accuracy (Section 4.1): The paper assumes that the ensemble of functions trained on human preferences accurately reflects the true objective. However, there's an inherent assumption that the datasets used encompass all aspects of "helpful and harmless" which might not be comprehensive. There is also a risk that the ensemble could inherit systemic biases or errors present in the training datasets.
2. Training the proxy model on a fixed dataset without further interaction with the gold model may not reflect the iterative nature of learning and adapting to new data.
3. There is a missing relevant work "TextGrad: Advancing Robustness Evaluation in NLP by Gradient-Driven Optimization, ICLR2023".
4. The paper notes a significant correlation of gamed examples across different proxy models. This suggests potential inherent weaknesses or biases in the training process or data. However, the paper does not provide a detailed analysis of the underlying causes of this correlation or strategies to mitigate it.
5. The appearance of phase shifts in reward curves during RLHF optimization highlights that current methods may not capture the complexity of policy learning dynamics. The paper's methods might be missing other, subtler forms of proxy gaming.

**Questions:**

I do not have other questions. Please refer to the weakness colum above.

---

> ### Author Response · Authors · 2023-11-22
> **Author Response**
>
> Thank you for your careful analysis of our work. We hope the following response addresses your concerns.
>
> **Quality of gold models.**
>
> The goal of our benchmark is to provide an automatic environment to study the proxy gaming problem. Using learning-based gold models instead of humans as evaluators greatly decreases cost and improves reproducibility. Furthermore, we take steps to ensure the “gold” models realistically reflect actual human preferences and the RLHF pipeline by 1) training on multiple standard human preference datasets, 2) using an ensemble of such models, and 3) only providing limited query access for training the proxy model.
>
> We cannot ensure that our datasets encompass all aspects of helpfulness and harmlessness, but this does not affect the validity of the evaluation. The gold models are fixed as ground-truth, following the methodology in Gao et al. (2022). This means that even though they are imperfect, the goal of the proxy model is to match the gold model, so the evaluation is fully valid. In this methodology, the gold model effectively stands in for a human labeler while enabling automatic evaluations.
>
> **Non-interaction with gold model.**
>
> > Training the proxy model on a fixed dataset without further interaction with the gold model may not reflect the iterative nature of learning and adapting to new data.
>
> We agree that this benchmark can also be used to investigate the dynamic of iterative and adaptive learning and we make our benchmark flexible enough to support such usage. While it is expected that active learning or simply training on more data points is one way of reducing proxy gaming effects, we establish a setting for users to limit query access to the gold model by using the provided comparison dataset for two reasons. 1) Querying humans (or the gold objective) is often expensive in practice. The proxy model is developed exactly due to such restrictions. By limiting query access to the gold model, users can focus on developing robustness techniques that work under resource constraints. 2) Having access to more query access to the gold model can give an unfair advantage for a technique, making comparisons more difficult. If one decides to query the gold model in addition to the comparison dataset, we encourage the user to clearly report such practices to facilitate fair comparison.
>
> **Choice of adversarial attacks.**
>
> PEZ and GBDA are representative of SoTA white-box attacks on text models, which are standard baselines in most papers. We agree that TextGrad is also a relevant attack. We have added it to the related work in the updated paper. Thank you for your suggestion.
>
>
> **Correlation of gamed examples across proxy models.**
>
> > The paper notes a significant correlation of gamed examples across different proxy models. This suggests potential inherent weaknesses or biases in the training process or data. However, the paper does not provide a detailed analysis of the underlying causes of this correlation or strategies to mitigate it.
>
> The correlation between gamed examples in the Base and Small proxy models is not itself an issue. It does demonstrate that some aspect of the data distribution is likely causing per-example proxy gaming, and that the gamed examples are not random. This is useful to know, but the issue is that proxy gaming happens at all. The goal of our paper and benchmark is to facilitate future work on developing more robust proxies to ameliorate this problem.

---

### Official Review · Reviewer_2dFK · 2023-11-05

**Soundness:** 2 fair
**Presentation:** 2 fair
**Contribution:** 2 fair
**Rating:** 5
**Confidence:** 3

**Summary:**

This paper introduces ProxyBench, a benchmark for investigating the robustness of neural network proxies under various sources of optimization pressure. Through extensive experiments, the authors uncover previously unknown properties of the proxy gaming problem and highlight serious issues with proxy reward models currently used to fine-tune or monitor large language models. They also find that common robustness enhancement techniques such as adversarial training provide limited gains for proxy robustness, and suggest new methods that may be required.

**Strengths:**

* This paper studies the proxy gaming problem in RLHF, which is an interesting and important problem.
* This paper conducted extensive experiments and revealed many previously unknown phenomena in proxy gaming
* The writing is overall clear and easy to follow

**Weaknesses:**

*  This paper mainly targets a benchmark for measuring proxy robustness. However, I believe most of the setup/data are borrowed from the existing works, in particular, [1], which weakens the originality.
*  This paper did introduce a novel metric to measure the severity of proxy gaming, namely the difference between the first derivatives of the proxy and golden reward curves as the optimization proceeds (RMSE). However, the authors didn't highlight the importance and necessity of such a metric. The major motivation from what I can see is the phenomenon that proxy gaming can still happen at a subset of the examples even if the expectation of the proxy reward aligns with the gold reward. But it is rather hard to see why the proposed metric can reflect the severity of this problem.
* Further, I am not sure why the aforementioned phenomenon, which is the major discovery of this paper, is important itself. I believe using the expected target across the data examples as a metric is rather a standard practice for probably any learning problem. Randomness always exists in the dataset and considering, for example, a worst-case target as a metric, is probably impractical.
* Nevertheless, such a phenomenon can indeed be important if the subset of examples experiencing serious proxy gaming, is relatively consistent across different proxy models. Because this points to some fundamental behavior changes of the policy model, which may lead to arbitrarily worse golden reward if the data distribution for evaluation is skewed towards the favor of the policy model. To clarify this, the authors did mention that this phenomenon may not be random at the end of Section 5.1, and that the policy model may exploit the flaw of the proxy in Section 5.4. However, there is no quantitative result mentioned, especially for the former, making it hard to judge whether they are indeed the case.

* Finally, I believe this is an interesting paper and has great potential. But it may not be well-organized in its current form. Personally, I would prefer pitching it as a comprehensive analysis of the proxy gaming problem, rather than as a benchmark. But certainly the former would require more extensive empirical results for more soundness.

* [Minor point] The related work is a little bit lacking and makes it quite difficult for people to catch up, as I believe the focused problem is brand new. The authors should at least introduce RLHF before reward modeling and proxy gaming.


[1] Scaling laws for reward model overoptimization. Gao et al., 2022.

**Questions:**

Please address the weaknesses mentioned above.

---

> ### Author Response · Authors · 2023-11-22
> **Author Response**
>
> Thank you for your careful analysis of our work. We hope the following response addresses your concerns.
>
> **Contributions relative to prior work.**
> There are three important shortcomings with the work of Gao et al. (2022) that we overcome in our paper: 1) they don’t release the code or the dataset which limits applicability, 2) they don’t introduce standardized metrics that reliably track progress, and 3) we have a series of qualitatively different results that are worthy of investigation (e.g., contrary to their findings, we observe larger policies can sometimes cause more severe proxy gaming, demonstrating adversarial effects in the RLHF setting).
>
> **Motivation for RMS metric.**
>
> > The major motivation from what I can see is the phenomenon that proxy gaming can still happen at a subset of the examples even if the expectation of the proxy reward aligns with the gold reward.
>
> This is not actually the motivation for our RMS metric. We do have a “% of Examples Gamed” metric for this, but the motivation for our RMS metric is to simply measure the average amount of proxy gaming across a dataset of examples. The closest prior work to ours, Gao et al. (2022), does not propose a quantitative metric for proxy gaming, so we developed a suitable metric for measuring proxy gaming based on the shape of reward curves.
>
> **Importance of per-example proxy gaming.**
>
> > I am not sure why the aforementioned phenomenon, which is the major discovery of this paper, is important itself.
>
> The importance of our discovery of per-example proxy gaming is that it demonstrates proxy gaming can still occur even when the dataset-level proxy reward curve tracks the dataset-level gold reward curve. This is concerning, because a standard choice when using RLHF is to simply monitor dataset-level statistics, especially because computing higher-quality gold rewards is assumed to be costly. One might naively assume that a slight mismatch between the dataset-level curves is due to a slight mismatch in the reward curves for most examples. Our results demonstrate that in fact, this is often due to severe proxy gaming on a subset of examples, which is concerning from a safety perspective.
>
> > I believe using the expected target across the data examples as a metric is rather a standard practice for probably any learning problem.
>
> Absolutely! Our RMS metric is a standard metric for measuring the difference in shape between two curves, and we average it over all the examples in our dataset. This measures the average amount of proxy gaming on a dataset level. The “% of Examples Gamed” metric is specifically motivated by our novel observation that subsets of examples can experience severe proxy gaming.
>
> **Clarifying the degree of per-example proxy gaming.**
>
> Our apologies for not being clear about the quantitative setup in Section 5.1. We observe that over 45% of the gamed examples in the Base proxy model also coincide with the gamed subset of examples in the Small proxy model. Running a simple hypothesis test compared to if the gamed examples were chosen at random, we find that there is a near zero probability of having 45% of the gamed examples overlap, thus rejecting the null hypothesis. We have clarified this in the updated paper thanks to your suggestion.
>
>
> **Reasons for developing a proxy gaming benchmark.**
>
> The primary goal of our paper is to facilitate the development of more robust proxy reward models. As shown by Gao et al. (2022), proxy gaming in RLHF is a serious problem. There are many possible research avenues for developing more robust proxy reward models, such as adversarial training and incorporating uncertainty. However, measuring proxy gaming can be challenging, and there is not yet any standardized evaluation for proxy gaming. For these reasons, we decided to create a standardized evaluation that would make it much easier for future work to explore ways of improving robustness.
>
> While our paper does include some interesting discoveries and analysis in addition to the benchmark, we view the benchmark as an important contribution that we hope will facilitate more work in this area.
>
>
> **Placement of RLHF in the related work.**
>
> The “Reward Modeling” paragraph at the top of our related work includes the standard InstructGPT RLHF paper as well as more recent methods like Constitutional AI. We assume familiarity with RLHF in our paper, so we don’t provide an explainer, but readers can easily link to the RLHF papers from our related work to learn more.

---

### Official Review · Reviewer_k5g4 · 2023-11-06

**Soundness:** 3 good
**Presentation:** 2 fair
**Contribution:** 3 good
**Rating:** 5
**Confidence:** 3

**Summary:**

This paper studies the proxy gaming phenomenon when we use a proxy model to learn policies. By introducing two evaluation metrics, final gold reward and RMS divergence, the authors obtain several empirical observations via benchmarking various training algorithms. In the end of the paper, the authors also provide several insights into designing methods to mitigate proxy gaming.

**Strengths:**

The proxy model is broadly used in policy training, especially in the case when evaluation is costly. It is necessary and interesting to study the phenomenon of proxy gaming, especially in the context of LLMs. In addition, the authors also provide several hints on how to mitigate proxy gaming in this context.

**Weaknesses:**

1. My major concern is the evaluation metric, since almost all conclusions of Section 5.2 are based on this metric, it is necessary to elaborate further why you chose this metric and the mathematical formulation of P' and G'. From my point of view, let $\\\{e_i\\\}_{i = 1}^T$ represent the difference of each instance between the proxy reward and gold reward, we can use the loss $(\frac{1}{T} \sum\_{i = 1}^T e\_i^\gamma)^{1 / \gamma}$ with a hyper-parameter $\gamma$ adjusting how much we should emphasis the instance-wise gap.

2. The title of this paper is a bit misleading. First, all the investigations are conducted in the text domain. Second, I think the key part of the paper is to benchmark the proxy gaming phenomenon under various settings. In addition to adversarial optimization (WB), BoN and reinforcement learning methods are also studied.

**Questions:**

My major concerns are pointed out in the "weakness" part, I welcome the authors to address my concerns. I will do a re-evaluation after receiving the authors' feedback. In addition, we have the following questions or confusion.

1. The gold objective here is also a learning-based model constructed from a few dataset. Have the authors evaluated how "gold" these models are? There is still a gap between the RLHF practice and the gold model used for analysis in this work. I am not sure how big this gap is.

2. Any intuition of the claim "This might be explained by a larger policy providing a stronger attack and finding more ways to exploit the proxy model." in the first part of Section 5.2?

3. In classification or regression problems, adversarial training may hurt the performance on the clean input, does the adversarial training technique introduced in Section 5.5 hurt the performance when the proxy model is not adversarial?

---

> ### Author Response · Authors · 2023-11-22
> **Author Response**
>
> Thank you for your careful analysis of our work. We hope the following response addresses your concerns.
>
> **Reasoning behind our RMS metric.**
>
> There are many possible metrics for quantifying the difference between two curves, such as the one you’re suggesting. However, the proxy gaming effect is mainly manifested in the reward curves having different shapes (which is better captured with second-order properties) instead of the “gap” between the reward curves. Consider a simple example R_gold = [4,2,2,1,1,1], R_proxy1 = [8,4,4,2,2,2], R_proxy2 = [0,1,2,2,2,3], where only R_proxy2 exhibits severe proxy gaming effects because the proxy reward increases while the gold reward decreases. Such relative directions can be better tracked by a metric that uses the second-order property. P’ = P[1:] - P[:-1].
>
>
> **Title of paper.**
>
> For the camera-ready version, we can change the title of the paper to “Robustness Evaluation of Text-Based Proxy Models Against Optimization Pressure”. Let us know if this looks good to you. Thank you for your suggestion. If we have addressed the thrust of your concerns, we kindly ask that you consider raising your score.
>
> **Responses to questions**
>
> > The gold objective here is also a learning-based model constructed from a few dataset. Have the authors evaluated how "gold" these models are?
>
> The goal of this benchmark is to provide an automatic environment to study the proxy gaming problem. Using learning-based gold models instead of humans as evaluators greatly decreases cost and improves reproducibility. Furthermore, we take steps to ensure the “gold” models realistically reflect actual human preferences and the RLHF pipeline by 1) training on multiple standard human preference datasets, 2) using an ensemble of such models, and 3) only providing limited query access for training the proxy model.
>
> > Any intuition of the claim "This might be explained by a larger policy providing a stronger attack and finding more ways to exploit the proxy model." in the first part of Section 5.2?
>
> Larger policy models afford a larger search space for RL, since there are more parameters to modify. Hence, they are stronger optimizers and more capable of overfitting to a proxy.
>
> > In classification or regression problems, adversarial training may hurt the performance on the clean input, does the adversarial training technique introduced in Section 5.5 hurt the performance when the proxy model is not adversarial?
>
> Exactly as you point out, we observe performance degradation of the adversarially trained models and report the accuracies in Table 4. The PEZ column refers to adversarial training using the PEZ optimizer. We have clarified this in the updated paper.

---

### Meta-Review · Area_Chair_RJHx · 2023-12-06

**Metareview:**

This paper introduces ProxyBench to rigorously assess the robustness of neural network proxies. The significance of the problem addressed in this paper has been unanimously acknowledged by all reviewers. Many reviewers also recognize the contribution of the paper for revealing many insightful conclusions via extensive experiments. However, there still exist common concerns from the reviewers regarding the golden reward (Reviewer k5g4 and tXHZ), the choice of evaluation metrics (Reviewer k5g4 and 2dFK), and the overall significance of this work (Reviewer 2dFK and 8v7i). Reviewer 2dFK maintains reservations about the paper's breadth of contribution, particularly noting a lack of comprehensive quantitative analyses. Additionally, the issues surrounding the 'golden reward' model remain unresolved. Considering the opinions of the reviewers and the overall rating, this paper is still worth further improvement and cannot be accepted.

**Justification For Why Not Higher Score:**

The paper did not receive a higher score due to several unresolved concerns from the reviewers. These include issues with the 'golden reward' model, questions about the lack of comprehensive quantitative analyses, and doubts about the breadth of the paper's contribution. These factors collectively suggest the need for further improvement in the paper.

**Justification For Why Not Lower Score:**

N/A

---

### Decision · Program_Chairs · 2024-01-16

Reject